# Learning Attribute–Affordance Hierarchies in Hyperbolic Space for Open-Vocabulary 3D Object Affordance Grounding

Yuxuan Wang [1]  Tong Li [1]  Yihang Zhu [1]  Guangtao Lyu [1]  Yukuan Min [1]  Chenghao Xu [2]  Jiexi Yan [1]  Xu Yang [1]
Cheng Deng [1] *

## Abstract

This paper pays attention to open-vocabulary 3D object affordance grounding (OVAG), which aims to localize affordance regions on 3D objects by leveraging interaction images or textual instructions. Most existing methods treat interaction images as sources of external affordance knowledge and align them with 3D visual representations, while overlooking the intrinsic relationship between local object attributes and affordances, which limits localization accuracy and generalization. For instance, a cup handle affords grasping due to its curved shape and appropriate thickness, indicating that affordances emerge from specific attribute compositions rather than global object appearance. Motivated by this, we propose Attribute-Affordance Hierarchies (AAH) learning framework that explicitly models the hierarchical relationships between object-region attributes and affordances. Our approach first captures local region relationships using hypergraph, and then projects these region-level concepts into a hyperbolic space to encode their hierarchical organization. Furthermore, we introduce counterfactual attribute samples to encourage robust learning of attribute–affordance dependencies under varying conditions. By jointly modeling visual structure and hierarchical concept information, our method achieves more accurate affordance localization. Extensive experiments and qualitative analyses demonstrate the effectiveness of our approach.

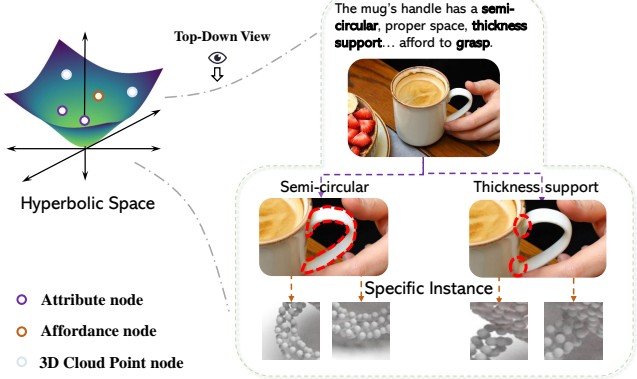

*Figure 1.* Modeling the semantic hierarchical structure. Affordance regions do not exist in isolation but are hierarchically related to local object attributes. To this end, we construct a semantic hierarchy that maps from attributes to affordance regions, capturing the abstract-to-specific process underlying affordance localization. Hyperbolic space naturally preserves such hierarchical structures: the origin corresponds to the most general concepts, while the distance from the origin encodes increasing semantic specificity.

## 1. Introduction

Affordance refers to the potential actions an object enables, which originates from ecological psychology (Gibson, 1979). Open-vocabulary 3D object affordance grounding (OVAG) aims to localize action-relevant regions on 3D objects—*i.e.*, where an object affords an interaction—under both seen and unseen scenarios (Shao et al., 2025; Li et al., 2024d). This problem is often cast as 3D affordance segmentation to identify affordance parts that bridge perception and operation for embodied agents, with strong impact on interaction systems (Geng et al., 2023; Rezapour Lakani et al., 2019; Moldovan et al., 2012) and broader applications such as robot manipulation (Huang et al., 2024; Kuang et al., 2024), scene understanding (Huang et al., 2023; Jia et al., 2024; Liu et al., 2024) and imitation learning (Hussein et al., 2017; Li et al., 2024b).

Training models to learn affordance knowledge is a challenging task. Most existing approaches (Yang et al., 2023;

---

[1]School of Electronic Engineering, Xidian University, Xian 710071, China [2]College of Information Science and Engineering, Hohai University, Changzhou 210024, China. Correspondence to: *Cheng Deng <chdeng.xd@gmail.com>.

*Proceedings of the 43rd International Conference on Machine Learning*, Seoul, South Korea. PMLR 306, 2026. Copyright 2026 by the author(s).

Nagarajan & Grauman, 2020) treat interaction images as sources of external affordance knowledge, aligning them with 3D visual representations to guide affordance localization. Related studies in cross-modal representation learning further emphasize the importance of preserving semantic consistency across heterogeneous modalities. For example, self-supervised adversarial hashing (Li et al., 2018) and triplet-based hashing (Deng et al., 2018) learn discriminative cross-modal representations by enforcing semantic alignment and relative similarity constraints, while dual-autoencoder-based spectral clustering (Yang et al., 2019) explores latent structural relationships among samples for more structured embedding learning. While effective to some extent, such visual-only settings make the segmentation models confined to visually predefined affordance types. Consequently, several studies (Li et al., 2024d; Van Vo et al., 2024) incorporate instructional cues, such as natural language descriptions, and align them with 3D geometries to inject semantic priors about object–interaction relationships, thereby partially alleviating the generalization gap caused by affordance diversity.

Despite their remarkable progress, these paradigms largely overlook the intrinsic relationship between local object attributes and affordances, which limits their localization accuracy. For instance, a cup handle affords grasping not merely because of observed interactions, but due to curved shape and appropriate thickness that enables stable grasping. Other methods primarily ground 3D affordances by aligning object geometry with instruction modalities. However, this strategy is highly dependent on the training data distribution. When encountering unseen affordances such as pour, without modeling the attribute-level information and their hierarchical relationship to affordances, models tend to rely on coarse visual similarity and collapse unseen affordances into frequently observed ones. This limitation underscores the necessity of capturing hierarchical relations between object-region attributes and affordances to enable robust generalization in open-vocabulary settings.

In this paper, we propose the AAH framework to model hierarchical relationships between object-region attributes and affordances, as illustrated in Figure 1. Given 3D object representations together with interaction images and textual prompts, our goal is to capture how affordances emerge from specific attribute compositions rather than from coarse object-level cues. To this end, we first employ a hypergraph-based module to model local interactions among object regions, providing a relation-aware foundation for affordance reasoning. Building upon these relational representations, we project region features into a hyperbolic space to encode the intrinsic attribute–affordance hierarchy, where semantics are naturally organized from abstract attributes to specific affordance regions. This hierarchical embedding enables more accurate reasoning about affordance regions. Further-

more, to improve robustness in open-vocabulary settings, we introduce counterfactual attribute prompts that impose structured perturbations on the learned hierarchy, encouraging the model to assess whether an affordance remains valid under altered attribute conditions. By jointly modeling visual structure and hierarchical concept information, our framework reduces reliance on spurious correlations and significantly improves generalization to unseen affordances. Extensive experiments and qualitative analyses validate the effectiveness of the proposed approach.

The contributions are summarized as follows:

(1) We propose to model the hierarchical relationships between object attributes and affordance using hyperbolic space, which enables more accurate localization in OVAG.

(2) We propose the AAH framework, which integrates hypergraph learning with hyperbolic space to capture attribute–affordance structures from local visual components.

(3) Experiments can demonstrate that AAH can localize affordance regions precisely, and yield state-of-the art results among multiple OVAG benchmarks.

## 2. Related Work

**Affordance Grounding.** Affordances (Gibson, 2014) are a key focus in robotics (Li et al., 2024c; Tang et al., 2025) and computer vision (Qian et al., 2024; Li et al., 2024a; Xu & Mu, 2025; Fang et al., 2018; Wang et al., 2025) because of their significant ability to connect perception with action. Typically, research on affordances in computer vision primarily focuses on 2D affordance detection and segmentation, aiming to identify objects or regions in images that support certain actions. These methods mainly rely on convolutional neural networks to infer affordance labels at the pixel or object level (Qian et al., 2024; Li et al., 2024a). Subsequent works introduce linguistic descriptions to augment affordance learning (Lu et al., 2022; Mi et al., 2020), yet most of them remain coarse-grained and focus on object-level affordances. To address this limitation, later studies shift towards part-level affordance learning, enabling more precise localization of interaction regions (Luo et al., 2022; Nagarajan et al., 2019). Despite their success, 2D affordance representations are inherently limited when deployed in real-world robotic systems, as they lack explicit 3D geometry and make it difficult to infer accurate interaction locations in physical space. This motivates the transition from 2D to 3D affordance grounding. Open-Vocabulary 3D Affordance Grounding (OVAG) aims to localize affordance regions under arbitrary and previously unseen semantic descriptions, posing a significant challenge beyond closed-set settings. Several recent works attempt to address this problem by leveraging cross-modal cues. IAGNet (Yang et al., 2023) utilizes 2D interaction semantics to guide 3D affor-

dance grounding, while LASO (Li et al., 2024d) introduces text-conditioned affordance queries and injects linguistic information into point features. OpenAD (Van Vo et al., 2024) further explore text–point correlations by exploiting large language–vision encoders such as CLIP to handle affordance synonym substitutions.

**Hyperbolic Representation Learning.** Hyperbolic representation learning has been widely adopted to model data with inherent hierarchical or tree-like structures, owing to its ability to embed such structures with low distortion (Nickel & Kiela, 2017; Chami et al., 2020). Following the seminal work on Hyperbolic Neural Networks (Ganea et al., 2018), hyperbolic geometry has been successfully integrated into a broad range of architectures, including convolutional networks (Shimizu et al., 2020), graph neural networks (Liu et al., 2019a) and more. Prior studies primarily interpret the hyperbolic radius from two perspectives: as a measure of prediction uncertainty (Ermolov et al., 2022; Franco et al., 2023), or as an explicit encoding of hierarchical parent–child relationships (Surís et al., 2021; Atigh et al., 2022). Our work exploits hyperbolic geometry to model the hierarchical dependency between object-region attributes and affordances at the region level. By embedding hypergraph-aggregated region features into a hyperbolic space, our approach captures coarse-to-fine affordance structures that naturally arise from attribute compositions.

## 3. Hypergraph and Hyperbolic Space

The OVAG task aims to localize affordance regions $Y_g$ on 3D object representation $P \in \mathbb{R}^{N \times 3}$ by leveraging knowledge learned from human–object interaction images $I$. During training, the supervision is limited to affordance annotation $P_{\text{label}} \in \mathbb{R}^{N \times 1}$. To address this challenging problem, we explore a combination of hypergraph and hyperbolic space to capture relationships among local visual components, which jointly models local features and hierarchical attribute–affordance associations, facilitating generalizable affordance grounding in open-vocabulary scenarios.

**Hypergraph.** A hypergraph can be defined as $\mathcal{G} = \{\mathcal{V}, \mathcal{E}, \mathcal{A}\}$, where $\mathcal{V}$ is a set of $\{v_1, \cdots, v_n\}$ vertices, $\mathcal{E}$ is a set of $\{e_1, \cdots, e_m\}$ hyperedges and $\mathcal{A} \in \{0,1\}^{\mathcal{V} \times \mathcal{E}}$ is an incidence matrix. $\mathcal{A}_{ij} = 1$ indicates that vertex $i$ belongs to hyperedge $j$. Hypergraph allows more than two vertices connect to a hyperedge, and more than two hyperedges connect to a vertex.

**Hyperbolic Space.** Hyperbolic space $\mathbb{H}^d$ is a Riemannian manifold with constant negative curvature, whose volume grows exponentially with the geodesic radius. This geometric property naturally aligns with tree-like and hierarchical data structures, alleviating the representational crowding commonly encountered in Euclidean embeddings and making hyperbolic geometry particularly suitable for modeling hierarchical relations such as attribute–affordance structures.

In this work, we adopt the Lorentz model, a realization of hyperbolic geometry that embeds $\mathbb{H}^d$ into the $(d+1)$-dimensional Minkowski space $\mathbb{R}^{1,d}$, thereby preserving geodesic distances and enabling closed-form and stable geometric operations. Specifically, for curvature $-c < 0$, the Lorentz model represents hyperbolic space as the upper sheet of a two-sheet hyperboloid:

$$\mathbb{H}_c^d = \left\{ \mathbf{u} \in \mathbb{R}^{d+1} : \langle \mathbf{u}, \mathbf{u} \rangle_{\mathcal{L}} = -\frac{1}{c}, u_{d+1} > 0 \right\}, \quad (1)$$

where $\langle \cdot, \cdot \rangle_{\mathcal{L}}$ denotes the Lorentzian inner product. For two vectors $\mathbf{u}, \mathbf{v} \in \mathbb{R}^{d+1}$ with spatial components $\tilde{\mathbf{u}}, \tilde{\mathbf{v}} \in \mathbb{R}^d$ and temporal components $u_{d+1}, v_{d+1} \in \mathbb{R}$, the Lorentzian inner product is defined as

$$\langle \mathbf{u}, \mathbf{v} \rangle_{\mathcal{L}} = \langle \tilde{\mathbf{u}}, \tilde{\mathbf{v}} \rangle_E - u_{d+1} v_{d+1}, \quad (2)$$

where $\langle \cdot, \cdot \rangle_E$ denotes the Euclidean inner product. For any two points $\mathbf{u}, \mathbf{v} \in \mathbb{H}_c^d$, the Lorentzian distance is defined as:

$$d_{\mathbb{H}}(\mathbf{u}, \mathbf{v}) = \frac{1}{\sqrt{c}} \operatorname{arccosh}\left( -c \langle \mathbf{u}, \mathbf{v} \rangle_{\mathcal{L}} \right). \quad (3)$$

The exponential map provides a principled mechanism for lifting Euclidean features into the hyperbolic manifold. We employ the exponential map (Li et al., 2025) centered at the hyperboloid origin $\mathbf{o} = (0, \ldots, 0, 1/\sqrt{c})$. Since any Euclidean feature vector $\mathbf{v} \in \mathbb{R}^d$ naturally resides in the tangent space $T_{\mathbf{o}} \mathbb{H}_c^d$, its temporal component is zero, which automatically satisfies the orthogonality condition $\langle \mathbf{o}, \mathbf{v} \rangle_{\mathcal{L}} = 0$.

## 4. Method

The core idea of our approach is to model the hierarchical relationships between object attributes and affordances, enabling more accurate localization of 3d affordance regions. To this end, we propose the AAH framework (see Figure 2). First, we employ a hypergraph to capture local relationships among visual components, providing a relation-aware geometric foundation for subsequent affordance understanding. Then, we incorporate counterfactual textual prompts to map the fused attribute, affordance, and 3D point cloud representations into a hyperbolic space that encodes hierarchical semantic relations. This design allows the model to learn more discriminative visual concepts and achieve more precise affordance region localization.

### 4.1. Hpergraph Construction

To focus the neural network on in-context local relation ships between different visual components, a hypergraph is constructed to estimate the potential correlations between feature points in semantic space. We first use DINO-ViT (Caron et al., 2021) to encode semantic information

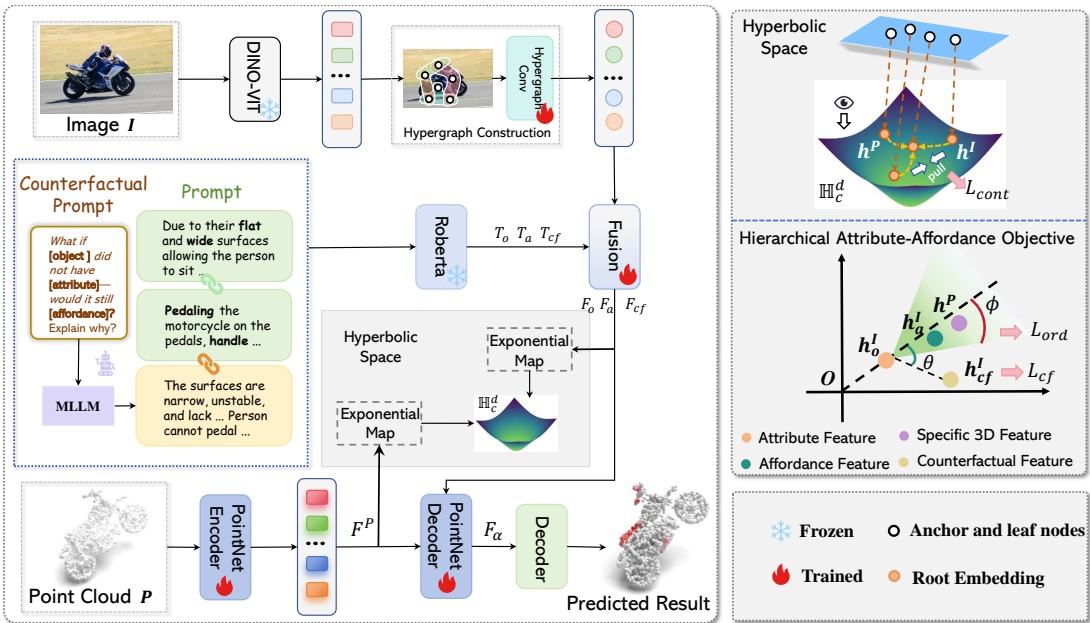

*Figure 2.* The details of AAH. Given 3D object representations together with interaction images and textual prompts, we first construct hypergraph to capture local relationships among object regions. These relational features are then projected into a hyperbolic space to explicitly encode the hierarchical structure between object-region attributes and affordances. In addition, counterfactual attribute prompts are introduced to refine the learned hierarchy. By modeling attribute–affordance hierarchies, the AAH enables more accuracy in OVAG.

$F \in \mathbb{R}^{C \times H \times W}$ from input RGB images $I$. Then, the feature maps $F$ are converted to hypergraph $\mathcal{G}$ to denote the semantic features' hypergraph structure. We view each feature vector within the feature map as a vertex and define the vertex sets $\mathcal{V}_{exo} = \{v_1, \cdots, v_{HW}\}$. To construct the hyperedge, a distance threshold $\epsilon$ is used to calculate an $\alpha$-space from vertex feature $v_i$ : $\alpha$-space $(v, \epsilon) = \{v_j \mid \|v_i - v_j\|_d < \epsilon, v_j \in \mathcal{V}\}$. The set of vertices present in this $\alpha$-space forms a hyperedge $e_i$. Every vertex within the feature map performs this operation. Then, the incidence matrix $\mathcal{A}$ is defined as:

$$\mathcal{A}(v, e) = \begin{cases} 1, & \text{if } v \in e \\ 0, & \text{otherwise.} \end{cases} \quad (4)$$

Based on this, the hypergraphs $\mathcal{G}$ are constructed. To facilitate message passing on the hypergraph structure, a hypergraph convolution (Feng et al., 2019) with residual connection is utilized to perform high order learning on vertex features $F$ follows:

$$\widetilde{F} = F + \boldsymbol{D}_v^{-1} \mathcal{A} \boldsymbol{D}_e^{-1} \mathcal{A}^\top F \Theta, \quad (5)$$

where $\boldsymbol{D}_v$ and $\boldsymbol{D}_e$ represent the diagonal degree matrices of the vertices and hyperedges, respectively. $\widetilde{F} \in \mathbb{R}^{C \times H \times W}$ indicate the relation enhanced features. The $\Theta$ is the learnable parameter. The hypergraph convolution promotes information exchange between vertices, enabling the model to better perceive in context local relationships.

## 4.2. Counterfactual Prompts Generation

Attribute and affordance-oriented prompts (Shao et al., 2025; Zhu et al., 2025) derived from interaction images provide strong semantic priors but often introduce spurious correlations, where frequently co-occurring attributes are incorrectly treated as necessary for affordance recognition, hindering generalization to unseen affordances and objects. To mitigate this issue, we introduce counterfactual attribute prompts on top of existing prompting frameworks. Given an interaction image $I$, we construct paired prompts consisting of a factual description and a counterfactual variant that perturbs the same attribute (*e.g.*, "What if the [object] did not have the [attribute], would it still afford the [affordance]? Explain why?"). A multimodal large language model (Chen et al., 2024) then leverages its world knowledge to reason about attribute–affordance relations' answer under counterfactual variations, reducing reliance on spurious cues and improving affordance region reasoning.

We then employ a RoBERTa (Liu et al., 2019b) text encoder to extract object attribute knowledge features $T_o \in \mathbb{R}^{N_o \times C}$, affordance intention knowledge features $T_a \in \mathbb{R}^{N_a \times C}$ and counterfactual attribute features $T_{cf} \in \mathbb{R}^{N_{cf} \times C}$. Where $N_o$, $N_a$ and $N_{cf}$ denote as the number of interaction objects, the number of interaction ways and the number of counterfactual conditions. To understand the multiple prompts of objects, the textual features are fused with the image

features as follows:

$$[F_o, F_a, F_{cf}] = f\left[\Gamma\left(T_o, T_a, T_{cf}\right), \widetilde{F}\right], \qquad (6)$$

where $\Gamma$ denotes reshape $T_o$, $T_a$, $T_{cf}$ to $\mathbb{R}^{C \times HW}$, $F_o, F_a, F_{cf} \in \mathbb{R}^{C \times HW}$. $f$ means two-layer Conv1d.

### 4.3. Hyperbolic Hierarchical Learning

After obtaining image features enriched with interaction intent, we embed these representations into a unified hyperbolic space to explicitly model the hierarchical relationships among image-level attributes, affordance cues, and 3D affordance regions. Further, the constraints are imposed to facilitate structured image-to-3D knowledge transfer, as illustrated in Figure 2.

**Point Feature Encode.** In addition to encoding image and textual inputs, the 3d point cloud are sent to PointNet++ (Qi et al., 2017). We use the output features from the final layer of the point cloud backbone as the point-wise representations $F_p \in \mathbb{R}^{C \times N_p}$.

**Hyperbolic Aggregation.** Given the image representations fused with multiple textual prompts and the 3D point cloud features, we first apply a geometry-aware weighting scheme based on the correlation between local tokens and global semantics. This design prevents the contributions of multiple local features from being over-smoothed during fusion, thereby preserving discriminative embedding capacity in the global representation. Let $F^I = \left\{F_o^k, F_a^k, F_{cf}^k\right\}_{k=1}^{HW}$ be fused image embeddings and $F^P = \left\{F_p^i\right\}_{i=1}^{HW}$ be point-wise features from the point cloud. We compute the initial anchor $(\bar{z}^I, \bar{z}^P) = \frac{1}{HW}\sum_{i=1}^{HW}(F^I, F^P)$ and using exponential map to map all anchors and leaf nodes to $\mathbb{H}_c^d$. For each leaf, we calculate its Lorentzian distance to the anchor as defined in Eq. 3. The resulting root embedding $h^I = \exp_o^c\left(\bar{z}^I\right)$, $h^P = \exp_o^c\left(\bar{z}^P\right)$. Then, we embed both modalities into a Lorentz-model hyperbolic space $\mathbb{H}_c^d$, where exponential volume growth naturally accommodates hierarchical organization. We then impose a Hierarchical Attribute-Affordance Objective that simultaneously (1) aligns cross-modal instances and (2) enforces the attribute-affordance ordering through entailment cones, and guides the model to learn which attributes remain sufficient to support affordance under the current conditions, thereby enabling effective generalization to unseen categories.

**Hierarchical Attribute-Affordance Objective.** To jointly achieve instance-level alignment and modeling of the attribute-affordance semantic hierarchy, we optimize a hierarchy-aware geometric objective defined entirely in hyperbolic space. The objective consists of a Positive Contrastive Loss for cross-modal alignment and an Attribute-Affordance Ordering Loss based on entailment cones, augmented with a Counterfactual Exclusion Loss to sharpen the hierarchy boundary.

*Positive Contrastive Loss.* Given a batch of prompt-conditioned image embeddings $\{h_i^I\}_{i=1}^B \in \mathbb{H}_c^d$ and corresponding point-cloud embeddings $\{h_j^P\}_{j=1}^B \in \mathbb{H}_c^d$, we define the similarity score using the Lorentz inner product:

$$s(i, j) = -d_{\mathbb{H}}(h_i^I, h_j^P)/\tau, \qquad (7)$$

where $\tau$ is a temperature parameter and $B$ is the batch size. The image-to-point cloud loss is formulated as:

$$L_{I \to P} = -\frac{1}{B}\sum_{i=1}^B \log \frac{\sum_{j \in P_i}\exp(s(i,j))}{\sum_{q=1}^B \exp(s(i,q))}. \qquad (8)$$

We symmetrize this objective by computing the corresponding point cloud-to-image loss $L_{P \to I}$, and define the final contrastive loss as:

$$L_{\text{cont}} = \tfrac{1}{2}\left(L_{I \to P} + L_{P \to I}\right). \qquad (9)$$

*Attribute-Affordance Ordering Loss.* To explicitly encode the semantic partial order that attributes entail affordances, we impose a nonsymmetric geometric constraint using entailment cones. For each attribute embedding $h_o^I \in \mathbb{H}_c^d$, we define a hyperbolic cone whose half-aperture contracts with increasing radius:

$$\phi(h_o^I) = \arcsin\left(\frac{2K}{\sqrt{c}\,\|\tilde{h}_o^I\|_E}\right), \qquad (10)$$

where $\left\|\tilde{h}_o^I\right\|_E$ denotes the spatial component of $h_o^I$, $c$ is the curvature, and $K = 0.1$ controls the maximal cone width. Given the affordance embedding $h_a^I$, we compute the exterior angle $\theta$ under Lorentz geometry. The ordering loss penalises those points that fall outside the cone as follows:

$$\begin{aligned} L_{\text{ord}}^{o \to a} &= \max\left(0,\ \theta(h_o^I, h_a^I) - \phi(h_o^I)\right), \\ L_{\text{ord}}^{a \to P} &= \max\left(0,\ \theta(h_a^I, h^P) - \phi(h_a^I)\right). \end{aligned} \qquad (11)$$

The first term enforces that the affordance embedding is semantically entailed by the corresponding attribute embedding. The second term propagates this hierarchy to the geometric level by constraining point-cloud features to fall inside the entailment cone defined by the affordance embedding, thereby grounding abstract affordance semantics into concrete 3D regions.

*Counterfactual Exclusion Loss.* To further refine the hierarchy, we introduce counterfactual attribute prompts that intentionally break the attribute-affordance relation. Let $h_{cf}^I$ denote the corresponding counterfactual embedding. We enforce that the true affordance embedding lies outside the counterfactual cone:

$$L_{\text{cf}} = \max\left(0,\ \phi(h_{cf}^I) - \theta(h_{cf}^I, h_a^I)\right). \qquad (12)$$

This exclusion mechanism prevents the model from relying on spurious attribute–affordance co-occurrences and encourages it to retain only those attributes that are truly necessary for the affordance.

### 4.4. Decoder and Total Loss

The image features enriched with interaction intent and the point features encoding geometric structure are jointly fed into the decoder to predict 3D affordance regions. Formally, we compute the affordance feature representation as:

$$\bar{F}_p = f[F_p, f_\Theta(T_o)], F_\alpha = f\big([f_\Gamma(F_a), Upsample(\bar{F}_p)]\big). \tag{13}$$

Concretely, to confuse $T_o \in \mathbb{R}^{N_o \times C}$ with the point cloud feature, we first choice $f_\Theta$, which expand $T_o$ to $\mathbb{R}^{N_p \times C}$. Then, we concatenate it with $F_p$ and use $f$, which indicates convolution layers with a 1×1 kernel, to encode them. After we get the fused feature $\bar{F}_p$, we follow the point cloud upsample operation to change the $\bar{F}_p$ to $\mathbb{R}^{C \times N}$ by Feature Propagation layers. $f_\Gamma$ reshapes the fused image features $F_a$ to $\mathbb{R}^{C \times N}$. And the final output $F_\alpha \in \mathbb{R}^{C \times N}$ is affordance feature representation. An output head $f_\phi(\cdot)$ followed by a sigmoid activation $\sigma(\cdot)$ produces the final affordance heatmap:

$$\phi = \sigma\big(f_\phi(F_\alpha)\big), \tag{14}$$

where $\phi \in \mathbb{R}^{N \times 1}$ denotes the predicted 3D object affordance. Then we directly supervise the discrepancy between the predicted affordance $\phi$ and the ground-truth affordance annotation $P_{label}$. Accordingly, the point cloud training objective consists of a Focal loss (Lin et al., 2017) and Dice loss (Milletari et al., 2016), both applied to point-wise affordance heatmaps:

$$L_{point} = L_{focal} + L_{dice}. \tag{15}$$

The overall training objective is a weighted combination of the above terms:

$$L_{\text{total}} = L_{point} + L_{\text{cont}} + \lambda\left(L_{\text{ord}}^{o \to a} + L_{\text{ord}}^{a \to P}\right) + \mu\, L_{\text{cf}}, \tag{16}$$

where $\lambda$ and $\mu$ balance instance alignment, hierarchical consistency, and counterfactual exclusion.

## 5. Experiment

### 5.1. Experimental Setting

**Dataset and Metrics.** In this paper, we evaluate our method on the Point Image Affordance Dataset v2 (PIADv2). PIADv2 consists of paired 2D human–object interaction images and 3D object point clouds, where point clouds are collected from 3DIR (Yang et al., 2024), 3D-AffordanceNet (Deng et al., 2021), and Objaverse (Deitke et al., 2023). The interaction images are sourced from

AGD20K (Luo et al., 2022), OpenImages (Krasin et al., 2017), and publicly licensed web data. In total, PIADv2 contains 15,213 images and 38,889 point clouds, covering 43 object categories and 24 affordance categories, making it the largest benchmark for 3D object affordance grounding to date. Each point cloud instance is densely annotated with point-wise affordance heatmaps, while images are labeled by affordance categories without requiring strict one-to-one correspondence with 3D instances. Following prior work, PIADv2 is evaluated under three settings: (1) Seen, where training and test sets share the same object and affordance categories; (2) Unseen Object, where affordances are shared but test objects are not observed during training; and (3) Unseen Affordance, where affordance categories and certain objects in the test set are excluded from training. For evaluation, we adopt commonly used metrics in recent 3D affordance grounding study (Shao et al., 2025), including Area Under Curve (AUC), average Intersection over Union (aIoU), SIMilarity (SIM), and Mean Absolute Error (MAE), to quantitatively measure the alignment between predicted and ground-truth affordance heatmaps.

**Implementation Details.** We take PointNet++ (Qi et al., 2017) and DINO-ViT-S (Caron et al., 2021) as the default 3D and 2D backbones, respectively. To ensure a fair comparison, the same feature extractor and settings are used to reproduce the baselines. We train the AAH for 65 epochs with a batch size of 16, utilizing the Adam optimizer with a learning rate of 1e-4. $\lambda = 1$ and $\mu = 1$. More details about the setting can be found in the appendix.

### 5.2. Results and Analysis

To provide a comprehensive comparison, we report quantitative results of our method against representative approaches for open-vocabulary 3D object affordance recognition. As shown in Table 1, our method consistently outperforms all competing baselines across all evaluation metrics under the three evaluation partitions, achieving state-of-the-art performance. Notably, our approach maintains stable advantages not only in the Seen setting but also in more challenging generalization scenarios, validating the effectiveness of the proposed open-vocabulary affordance recognition framework. The qualitative results in Fig. 4 further support these findings. While most methods exhibit comparable performance in the Seen setting, clear differences emerge in the Unseen scenarios. For instance, in the support and lay examples under the Unseen setting, methods relying solely on textual cues, visual cues, or direct alignment between object geometry and textual descriptions fail to generalize effectively. In contrast, our method leverages hyperbolic-space-based reasoning to uncover fine-grained interaction cues from 2D human–object interaction images, resulting in more accurate affordance predictions. More visualization results can be found in appendix.

*Table 1.* **Comparison on the PIADv2.** Evaluation metrics of comparison methods on the benchmark, the best results are in **bold**. Seen, Unseen Object and Unseen Affordance are three partitions of the dataset. AUC and aIOU are shown in percentage. Underlined results "‾" correspond to replacing the ResNet backbone with DINO-ViT-S, while keeping all other settings unchanged.

| Methods | Seen | | | | Unseen Object | | | | Unseen Affordance | | | |
|---|---|---|---|---|---|---|---|---|---|---|---|---|
| | AUC ↑ | aIOU ↑ | SIM ↑ | MAE↓ | AUC ↑ | aIOU ↑ | SIM ↑ | MAE↓ | AUC ↑ | aIOU ↑ | SIM ↑ | MAE↓ |
| Baseline | 87.04 | 34.18 | 0.594 | 0.079 | 72.74 | 16.34 | 0.336 | 0.156 | 58.09 | 7.88 | 0.208 | 0.160 |
| FRCNN | 87.05 | 33.55 | 0.600 | 0.082 | 72.20 | 18.08 | 0.362 | 0.152 | 59.08 | 7.96 | 0.210 | 0.156 |
| XMF | 87.39 | 33.91 | 0.604 | 0.078 | 74.61 | 17.40 | 0.361 | 0.126 | 60.99 | 8.11 | 0.225 | 0.152 |
| IAG | 89.03 | 34.29 | 0.623 | 0.076 | 73.03 | 16.78 | 0.351 | 0.123 | 62.29 | 8.99 | 0.251 | 0.141 |
| LASO | 90.34 | 34.88 | 0.627 | 0.077 | 73.32 | 16.05 | 0.354 | 0.123 | 64.07 | 8.37 | 0.228 | 0.140 |
| GREAT | 91.99 | 38.03 | 0.676 | 0.067 | 79.57 | 20.16 | 0.402 | 0.109 | 69.81 | 12.05 | 0.290 | 0.127 |
| G̲R̲E̲A̲T̲ | 91.95 | 38.01 | 0.671 | 0.066 | 79.59 | 20.19 | 0.403 | 0.109 | 69.82 | 12.06 | 0.291 | 0.126 |
| **Ours** | **93.29** | **38.98** | **0.688** | **0.060** | **81.01** | **21.18** | **0.430** | **0.090** | **70.90** | **13.26** | **0.320** | **0.113** |

## 5.3. Ablation Study

**Ablation Study of Distance Threshold $\epsilon$.** We validate the effectiveness of the distance threshold $\epsilon$ used in the construction of the hypergraph, with the results shown in Figure 3. We show the AUC evaluation metric results, where higher is better. It is observed that the performance of the 3d point cloud localization network improves across a range of threshold values from 1 to 3. However, there is a performance decline at the thresholds of 4 and 5. This performance degradation can be attributed to the influence of vertex connectivity on feature smoothness within the affordance semantic space. A larger distance threshold produces a more densely connected hypergraph, encouraging excessive information sharing among vertices and consequently introducing background features that are irrelevant to the interaction region. In contrast, an overly small threshold results in sparsely connected hyperedges, limiting the model's ability to capture meaningful relationships among affordance-relevant visual components. Based on this trade-off, we empirically set the distance threshold to 3, which provides a balanced level of connectivity for effective affordance reasoning.

**Ablation Study of Loss Function and Hypergraph.** We analyze the impact of each component in Table 2. Removing $L_{\text{cont}}$ causes the most severe drop, as ordering constraints alone cannot guarantee sufficient instance-level discrimination. This is because ordering constraints alone are insufficient to enforce discriminative instance-level representations, highlighting the necessity of jointly combining contrastive alignment with hierarchical constraints to achieve both semantic separation and structured organization. Removing $L_{\text{ord}}$ leads to a clear performance drop, particularly in Unseen settings. This observation indicates that without hierarchical supervision, the model struggles to preserve the intended attribute–affordance structure and tends to collapse concepts and instance-specific 3d semantics into a flat sim-

*Table 2.* **Ablation studies.** Performance when not modeling our method with hypergraph (Hyper.), loss $L_{\text{cont}}$, $L_{\text{ord}}$ and $L_{\text{cf}}$. ✗ means without.

| | Metrics | Ours | ✗ Hyper. | ✗ $L_{\text{cont}}$ | ✗ $L_{\text{ord}}$ | ✗ $L_{\text{cf}}$ |
|---|---|---|---|---|---|---|
| **Seen** | AUC | 93.29 | 92.88 | 92.00 | 92.65 | 92.75 |
| | aIOU | 38.98 | 38.71 | 38.05 | 38.50 | 38.65 |
| | SIM | 0.688 | 0.682 | 0.672 | 0.679 | 0.681 |
| | MAE | 0.060 | 0.062 | 0.066 | 0.063 | 0.062 |
| **Unseen Object** | AUC | 81.01 | 80.71 | 79.65 | 80.55 | 80.60 |
| | aIOU | 21.18 | 20.98 | 20.25 | 20.69 | 20.85 |
| | SIM | 0.430 | 0.425 | 0.405 | 0.417 | 0.423 |
| | MAE | 0.090 | 0.093 | 0.109 | 0.101 | 0.095 |
| **Unseen Affordance** | AUC | 70.90 | 70.40 | 69.85 | 70.10 | 70.30 |
| | aIOU | 13.26 | 12.96 | 12.10 | 12.45 | 12.85 |
| | SIM | 0.320 | 0.300 | 0.292 | 0.296 | 0.299 |
| | MAE | 0.113 | 0.115 | 0.126 | 0.120 | 0.117 |

ilarity space, thereby impairing generalization. Removing the counterfactual loss $L_{\text{cf}}$ results in a modest performance drop, which further confirms that counterfactual prompts are effective in suppressing spurious attribute–affordance correlations and guiding the model to identify which attributes are sufficient to support a given affordance under specific conditions. When simply using the backbone network (Removing Hyper.), the structure information will be neglected from the input, which leads to poor performance. This indicates that the hypergraph is an essential component.

**Ablation Study of Hyperbolic Space.** To verify the importance of hyperbolic geometry in modeling attribute–affordance hierarchies, we compare our model with two Euclidean-based variants, with results summarized in Table 3. The first baseline, EU+MP, replaces the hyperbolic embedding with a Euclidean space and adopts mean pooling for static feature fusion. This variant consistently yields the weakest performance across all settings, indicating that naive pooling in Euclidean space suffers from severely limited discriminability. We then consider EU+CL,

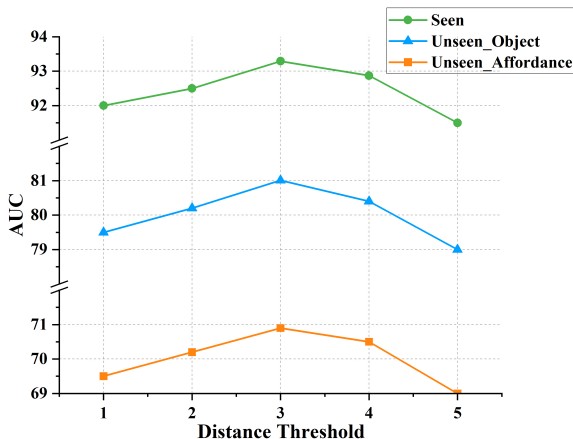

*Figure 3.* Ablation study on the distance threshold $\epsilon$. The results are shown on the PIADv2 test sets.

*Table 3.* **Ablation studies.** The effect of the hyperbolic space. "EU+MP" means pooling for static fusion, "EU+CL" means feature concatenation with linear layers.

| | Metrics | Ours | EU+MP | EU+CL |
|---|---|---|---|---|
| **Seen** | AUC | 93.29 | 92.20 | 92.75 |
| | aIOU | 38.98 | 38.10 | 38.55 |
| | SIM | 0.688 | 0.673 | 0.680 |
| | MAE | 0.060 | 0.066 | 0.063 |
| **Unseen Object** | AUC | 81.01 | 79.85 | 80.40 |
| | aIOU | 21.18 | 20.30 | 20.45 |
| | SIM | 0.430 | 0.408 | 0.411 |
| | MAE | 0.090 | 0.108 | 0.099 |
| **Unseen Affordance** | AUC | 70.90 | 70.00 | 70.25 |
| | aIOU | 13.26 | 12.20 | 12.35 |
| | SIM | 0.320 | 0.292 | 0.304 |
| | MAE | 0.113 | 0.125 | 0.121 |

*Table 4.* **Ablation studies.** The effect of the contrastive loss. We add the contrastive loss $L_{\text{cont}}$ to GREAT method for matched image-point cloud alignment.

| | Metrics | GREAT | GREAT + $L_{\text{cont}}$ |
|---|---|---|---|
| **Seen** | AUC | 91.99 | 92.34 |
| | aIOU | 38.03 | 38.20 |
| | SIM | 0.676 | 0.680 |
| | MAE | 0.067 | 0.064 |
| **Unseen Object** | AUC | 79.57 | 79.99 |
| | aIOU | 20.16 | 20.25 |
| | SIM | 0.402 | 0.411 |
| | MAE | 0.109 | 0.105 |
| **Unseen Affordance** | AUC | 69.81 | 70.10 |
| | aIOU | 12.05 | 12.22 |
| | SIM | 0.290 | 0.293 |
| | MAE | 0.127 | 0.125 |

*Table 5.* **Ablation studies.** The effect of the hyperbolic space. "EU+Hierarch." refers to the Euclidean baseline equipped with hierarchical supervision, "Hyperbolic+Standard" denotes a hyperbolic structure trained with the standard segmentation loss.

| | Metrics | Ours | EU+Hierarch. | Hyperbolic+Standard |
|---|---|---|---|---|
| **Seen** | AUC | 93.29 | 92.75 | 92.50 |
| | aIOU | 38.98 | 38.55 | 38.35 |
| | SIM | 0.688 | 0.680 | 0.677 |
| | MAE | 0.060 | 0.063 | 0.064 |
| **Unseen Object** | AUC | 81.01 | 80.40 | 80.32 |
| | aIOU | 21.18 | 20.45 | 20.36 |
| | SIM | 0.430 | 0.411 | 0.409 |
| | MAE | 0.090 | 0.099 | 0.101 |
| **Unseen Affordance** | AUC | 70.90 | 70.25 | 70.05 |
| | aIOU | 13.26 | 12.35 | 12.20 |
| | SIM | 0.320 | 0.304 | 0.298 |
| | MAE | 0.113 | 0.121 | 0.123 |

which concatenates features followed by linear layers for fusion. This variant achieves moderate improvements over EU+MP, confirming that stronger fusion mechanisms partially alleviate the limitations of Euclidean representations. Our full model, which embeds region features into a hyperbolic space and encodes hierarchical attribute–affordance relations, achieves the best performance across all metrics and evaluation partitions. This progressive improvement from EU+MP to EU+CL and ultimately to our full model directly validates the necessity and effectiveness of hyperbolic space for open-vocabulary 3D affordance grounding.

**Ablation Study of Contrastive Objective.** The contrastive loss term is primarily responsible for aligning matched image/point cloud instances. In contrast, the hyperbolic formulation further enforces semantic ordering through $L_{\text{ord}}$ and enhances the clarity of hierarchical boundaries via $L_{\text{cf}}$. To further investigate this, we incorporate the contrastive objective into the GREAT model for evaluation. The re-

sults, as shown in Table 4, indicate that the performance improvement is marginal. This suggests that contrastive alignment alone is insufficient to fully capture the hierarchical attribute-affordance relations required for OVAG, and that such contrastive objectives are more effective when coupled with the proposed hyperbolic structured representation.

**Ablation Study of Geometry and Supervision.** To disentangle the effect of geometry and supervision, we conduct additional controlled comparisons (Euclidean + hierarchical loss vs. Ours vs. Hyperbolic + hierarchical loss) on Table 5. While hierarchical supervision improves Euclidean models, the hyperbolic variant (Ours) consistently achieves better performance. This suggests that hyperbolic geometry and supervision provide a more suitable structure for modeling attribute-affordance-3d regions relations, rather than the gain being solely attributable to stronger loss design or feature fusion.

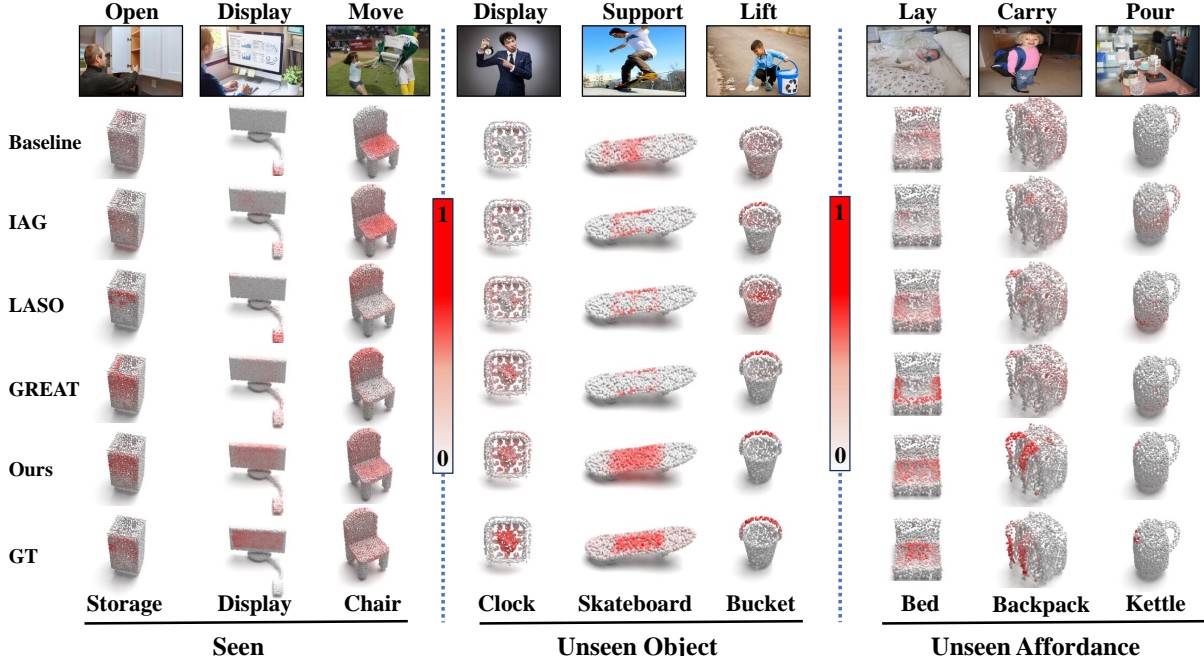

*Figure 4.* Visualization Results. The first row shows the interaction images, while the last row presents the ground-truth 3D object affordance annotations on point clouds. The left, middle, and right columns correspond to qualitative comparisons under the Seen, Unseen Object, and Unseen Affordance settings, respectively. The intensity of the red color indicates the predicted affordance probability.

## 6. Conclusion

We present AAH, a framework for OVAG that explicitly models hierarchical relationships between object-region attributes and affordances. By combining hypergraph-based relational modeling with hyperbolic hierarchical representations, our approach captures how affordances reason from attribute compositions and improves affordance region localization. Extensive experiments demonstrate the effectiveness of our method, particularly in challenging open-vocabulary scenarios. Our approach relies on textual prompts to specify attribute–affordance relations, and ambiguous or incomplete descriptions may weaken the constructed hierarchy in open-vocabulary settings. The use of hyperbolic entailment constraints introduces extra geometric computation, potentially affecting scalability to very large point clouds. Finally, the counterfactual loss depends on manually designed counterfactual prompts, and automatically generating diverse and reliable counterfactual conditions remains an open problem.

## Acknowledgments

Our work is supported in part by the National Key R&D Program of China (No. 2023YFC3305600) and the National Natural Science Foundation of China (U25B2048, 62132016).

## Impact Statement

This paper presents work whose goal is to advance the field of Machine Learning. There are many potential societal consequences of our work, none which we feel must be specifically highlighted here.

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

## A. Implementation Details

We summarize the dimensions and meanings of key tensors in our pipeline in Table 6. For the image branch, we adopt DINO-ViT-S as the feature extractor. The input image is randomly cropped and resized to $224 \times 224$, producing image features of $F \in \mathbb{R}^{512 \times 14 \times 14}$. A $1 \times 1$ convolution is then applied to project features to the desired embedding space, then we flatten the spatial dimensions to obtain $F \in \mathbb{R}^{512 \times 196}$. For the point branch, each input point cloud contains 2048 points. We employ PointNet++, which consists of three set abstraction (SA) layers to progressively extract multi-scale point features. In each SA layer, Farthest Point Sampling (FPS) is used with sampling counts set to 512, 128, and 64. The final point features are represented as $F_p \in \mathbb{R}^{512 \times 2048}$. To ensure its positivity, the curvature parameter $c$ is parameterized via its logarithm, i.e., the model learns $log(c)$, and is initialized to $c = 1.0$. The temperature $\tau$ in the contrastive loss is set to 0.07. We follow the GREAT method to get the prompt one to four. The prompt five is counterfactual attribute prompts. Detailed prompts used are listed below.

- **Prompt One:** *"Point out which part of the object in the image interacts with the person. If this part is different from the part of the object shown in the image that performs the main function, point out the part of the object that performs the main function shown in the image."*

- **Prompt Two:** *"Explain why this part can interact from the geometric structure of the object. Just give the final result in one sentence."*

- **Prompt Three:** *"Describe the interaction between object and the person in the image, including the interaction. . . "*

- **Prompt Four:** *"List two interactions that describe additional common interactions that the object can interact with people, including the interaction type, the interaction part of the object, and the interaction part of the person. . . "*

- **Prompt Five:** *"Explain why this part would fail to interact if its geometric structure of the {object} no longer satisfied the requirements for interaction. Just give the final result in one sentence. . . "*

We concatenate the answers of Prompt One and Prompt Two to form the object geometric knowledge, concatenate the answers of Prompt Three and Prompt Four to form the affordance intention knowledge, and use Prompt Five to obtain a counterfactual attribute description. The textual descriptions from Prompts One–Five are encoded by a RoBERTa text encoder to obtain the object geometric knowledge feature $T_o \in \mathbb{R}^{1 \times 512}$, the affordance intention knowledge feature $T_a \in \mathbb{R}^{3 \times 512}$, and the counterfactual attribute feature $T_{cf} \in \mathbb{R}^{1 \times 512}$.

*Table 6.* **Tensors.** The dimension and meaning of the tensors in the pipeline.

| Tensor | Dimension | Meaning |
|:---:|:---:|:---|
| $F$ | $512 \times 14 \times 14$ | image extractor output |
| $F_p$ | $512 \times 64$ | point cloud extractor output |
| $T_o$ | $1 \times 512$ | object geometric knowledge feature |
| $T_a$ | $3 \times 512$ | affordance intention knowledge feature |
| $T_{cf}$ | $1 \times 512$ | counterfactual attribute feature |
| $F_\alpha$ | $512 \times 2048$ | affordance feature representation |
| $\phi$ | $2048 \times 1$ | 3D object affordance |

**More Formulation Explanation.** (**Exponential map**) Let $\mathbb{H}_c^d$ denote the $d$-dimensional hyperbolic space with curvature $c$ under the hyperboloid model. We denote the Lorentz inner product by $\langle \cdot, \cdot \rangle_{\mathcal{L}}$. The tangent space at $w$ is $T_w\mathbb{H}_c^d = \{ v \in \mathbb{R}^{d+1} : \langle v, w \rangle_{\mathcal{L}} = 0 \}$, i.e., the $d$-dimensional subspace orthogonal to $w$ under $\langle \cdot, \cdot \rangle_{\mathcal{L}}$. For a point $w \in \mathbb{H}_c^d$ and a tangent vector $v \in T_w\mathbb{H}_c^d$, the Lorentz exponential map is

$$\exp_w^c(v) = \cosh\left(\sqrt{c}\,\|v\|_{\mathcal{L}}\right) w + \frac{\sinh\left(\sqrt{c}\,\|v\|_{\mathcal{L}}\right)}{\sqrt{c}\,\|v\|_{\mathcal{L}}}\, v, \tag{17}$$

where $\|v\|_{\mathcal{L}} = \sqrt{\langle v, v \rangle_{\mathcal{L}}}$ is the induced Lorentzian norm. In practice, we use the origin $o = (0, \ldots, 0, 1/\sqrt{c})$ and substitute $w = o$ to obtain $u = \exp_o^c(v)$, whose closed form is

$$\tilde{u} = \frac{\sinh\left(\sqrt{c}\,\|v\|_E\right)}{\sqrt{c}\,\|v\|_E}\, v, \qquad u_{d+1} = \frac{\cosh\left(\sqrt{c}\,\|v\|_E\right)}{\sqrt{c}}. \tag{18}$$

Here, $u \in \mathbb{R}^{d+1}$ is the embedded point on the hyperboloid, where $\tilde{u}$ denotes the spatial component (the first $d$ coordinates) and $u_{d+1}$ is the time coordinate (the last dimension). Eq. (18) is obtained by specializing Eq. (17) at $w = o$, where the spatial part scales $v$ by a hyperbolic factor and the time coordinate is determined by the curvature $c$, thereby faithfully placing the feature on the constant-curvature manifold and facilitating geometry-aware learning.

**(Exterior angle)** Given a paired 3D embedding $\mathbf{h}_p$, we compute the exterior angle between the cone axis $\mathbf{h}_t$ and $\mathbf{h}_p$ as

$$\theta(\mathbf{h}_t, \mathbf{h}_p) = \arccos\left(\frac{h_{p,d+1} + c \cdot h_{t,d+1}\,\langle \mathbf{h}_t, \mathbf{h}_p \rangle_{\mathcal{L}}}{\|\tilde{\mathbf{h}}_t\|_E\,\sqrt{\left(c\langle \mathbf{h}_t, \mathbf{h}_p \rangle_{\mathcal{L}}\right)^2 - 1}}\right), \tag{19}$$

where $\tilde{\mathbf{h}}_t$ denotes the spatial component of $\mathbf{h}_t$, $h_{t,d+1}$ and $h_{p,d+1}$ are the time coordinates of $\mathbf{h}_t$ and $\mathbf{h}_p$, respectively, and $\langle \cdot, \cdot \rangle_{\mathcal{L}}$ is the Lorentz inner product.

**Compared Baselines.** For a comprehensive comparison, we include two representative 3D affordance grounding methods, IAG (Yang et al., 2023) and LASO (Li et al., 2024d), as well as two image–point cloud cross-modal baselines, FRCNN (Xu et al., 2023) and XMF (Aiello et al., 2022), following the baseline suite used in IAG. We also compare with GREAT, a recent open-vocabulary 3D object affordance grounding method. Following IAG, all baselines use modality-specific feature extractors, and the fused representation is obtained by directly concatenating features from different modalities before the prediction head.

**Computational Complexity.** We compare our method with several existing approaches in terms of parameter count and computational cost, as shown in Table 7. The results show that our method introduces a moderate increase in both parameter count and inference time compared to the strongest baseline. This overhead mainly comes from the hypergraph module and the additional hyperbolic embedding and loss computations. However, these components operate on compact feature representations, and the overall increase remains limited relative to the backbone cost.

*Table 7.* **Statistics of time, model parameters and trainable parameters.**

| Model | Time | Model Parameters | Trainable Parameters |
|---|---|---|---|
| IAG | 1.426s | 24.7M | 24.7M |
| LASO | 1.336s | 130.5M | 130.5M |
| GREAT | 1.272s | 256.7M | 23.1M |
| Ours | 2.1s | 415.2M | 150M |

# B. More Comparative Experiments

To ensure a fair comparison, we train and evaluate our method on PIAD and compare it with IAG and GREAT, as reported in Table 8. We re-train all baselines on PIAD using the same training split and evaluation protocol. Our method achieves consistent improvements over both IAG and GREAT across seen and unseen settings.

*Table 8.* **Comparison on the PIAD.** Evaluation metrics of comparison methods on the PIAD benchmark, $\diamond$ denotes the relative improvement of our method over IAG method and GREAT method.

| Methods | Seen | | | | Unseen Object | | | | Unseen Affordance | | | |
|---|---|---|---|---|---|---|---|---|---|---|---|---|
| | AUC↑ | aIOU↑ | SIM↑ | MAE↓ | AUC↑ | aIOU↑ | SIM↑ | MAE↓ | AUC↑ | aIOU↑ | SIM↑ | MAE↓ |
| IAG | 82.88 | 18.88 | 0.544 | 0.098 | 68.49 ◇2.5% | 7.22 ◇4.7% | 0.344 ◇5.5% | 0.139 ◇12.2% | 55.36 ◇15.9% | 6.50 ◇4.0% | 0.203 ◇39.4% | 0.170 ◇20.6% |
| GREAT | 85.22 | **19.61** | 0.569 | 0.093 | 69.41 ◇1.2% | 7.49 ◇0.9% | 0.352 ◇3.1% | 0.127 ◇3.9% | 62.59 ◇2.5% | 6.56 ◇3.0% | 0.264 ◇7.2% | 0.143 ◇5.6% |
| Ours | **86.97** | 17.24 | **0.575** | **0.091** | **70.22** | **7.56** | **0.363** | **0.122** | **64.16** | **6.76** | **0.283** | **0.135** |

Table 9 reports the results of our method with different 3D backbones, including DGCNN and Point Transformer, on the PIADv2 benchmark. Compared with GREAT using the same backbone, our method consistently achieves better performance across the seen, unseen object, and unseen affordance splits. These results demonstrate that the proposed framework is not tied to a specific 3D encoder. Instead, it can be effectively integrated with both graph-based and transformer-based point cloud backbones. The consistent gains on the unseen object and unseen affordance splits further indicate that our hierarchical hyperbolic modeling improves generalization to novel object categories and unseen affordance concepts. This suggests that the performance improvement mainly comes from the proposed attribute-affordance region structure modeling, rather than from a particular backbone architecture.

*Table 9.* **Point Transformer and DGCNN of our method on the PIADv2.**

| | Seen | | | | Unseen Object | | | | Unseen Affordance | | | |
|---|---|---|---|---|---|---|---|---|---|---|---|---|
| **Methods** | **AUC↑** | **aIOU↑** | **SIM↑** | **MAE↓** | **AUC↑** | **aIOU↑** | **SIM↑** | **MAE↓** | **AUC↑** | **aIOU↑** | **SIM↑** | **MAE↓** |
| GREAT (DGCNN) | 92.32 | 38.22 | 0.679 | 0.065 | 79.86 | 20.42 | 0.409 | 0.108 | 69.93 | 12.07 | 0.296 | 0.125 |
| Ours (DGCNN) | **93.70** | **39.00** | **0.689** | **0.060** | **81.29** | **21.39** | **0.431** | **0.090** | **70.96** | **13.30** | **0.326** | **0.112** |
| GREAT (Transformer) | 92.91 | 38.64 | 0.682 | 0.063 | 80.15 | 20.66 | 0.412 | 0.107 | 70.24 | 12.10 | 0.299 | 0.123 |
| Ours (Transformer) | **94.20** | **39.23** | **0.690** | **0.057** | **81.67** | **21.65** | **0.433** | **0.087** | **80.11** | **13.46** | **0.328** | **0.111** |

## C. Prompt Sensitivity and Robustness

To investigate the robustness of our method to the quality of prompt information, we conduct noise injection experiments with noise ratios of 12.5% and 25%. As shown in Table 10, we observe that the model does not suffer from significant performance degradation and maintains competitive accuracy even under high noise conditions. We attribute this robustness to the joint effect of the counterfactual exclusion loss. This training strategy teaches the model to decouple affordance from auxiliary information. Consequently, at inference time, the model naturally learns to ignore irrelevant or noisy parts of the prompt that do not contribute to the essential causal chain of the affordance.

*Table 10.* **Noise robustness of our method on the PIADv2.**

| | Seen | | | | Unseen Object | | | | Unseen Affordance | | | |
|---|---|---|---|---|---|---|---|---|---|---|---|---|
| **Noise Rate** | **AUC↑** | **aIOU↑** | **SIM↑** | **MAE↓** | **AUC↑** | **aIOU↑** | **SIM↑** | **MAE↓** | **AUC↑** | **aIOU↑** | **SIM↑** | **MAE↓** |
| 0% | 93.29 | 38.98 | 0.688 | 0.060 | 81.01 | 21.18 | 0.430 | 0.090 | 70.90 | 13.26 | 0.320 | 0.113 |
| 12.5% | 93.02 | 38.71 | 0.683 | 0.061 | 80.63 | 20.86 | 0.423 | 0.093 | 70.41 | 12.98 | 0.312 | 0.117 |
| 25% | 92.63 | 38.32 | 0.676 | 0.064 | 80.14 | 20.44 | 0.414 | 0.097 | 69.82 | 12.61 | 0.303 | 0.121 |

## D. More Visual Results

We present qualitative visualizations of our method on three evaluation settings. Figure.6 shows representative results for the seen, unseen-object, and unseen-affordance splits. Overall, our method localizes affordance-relevant regions accurately in diverse interaction images and across multiple object categories, demonstrating strong robustness and generalization. In addition, we also show the heatmap comparison between the baseline and our method in Figure 5.

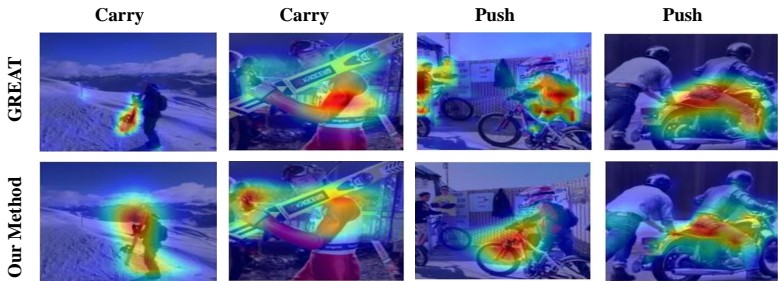

*Figure 5.* The first row are the results from the GREAT method, the last rows are our method.

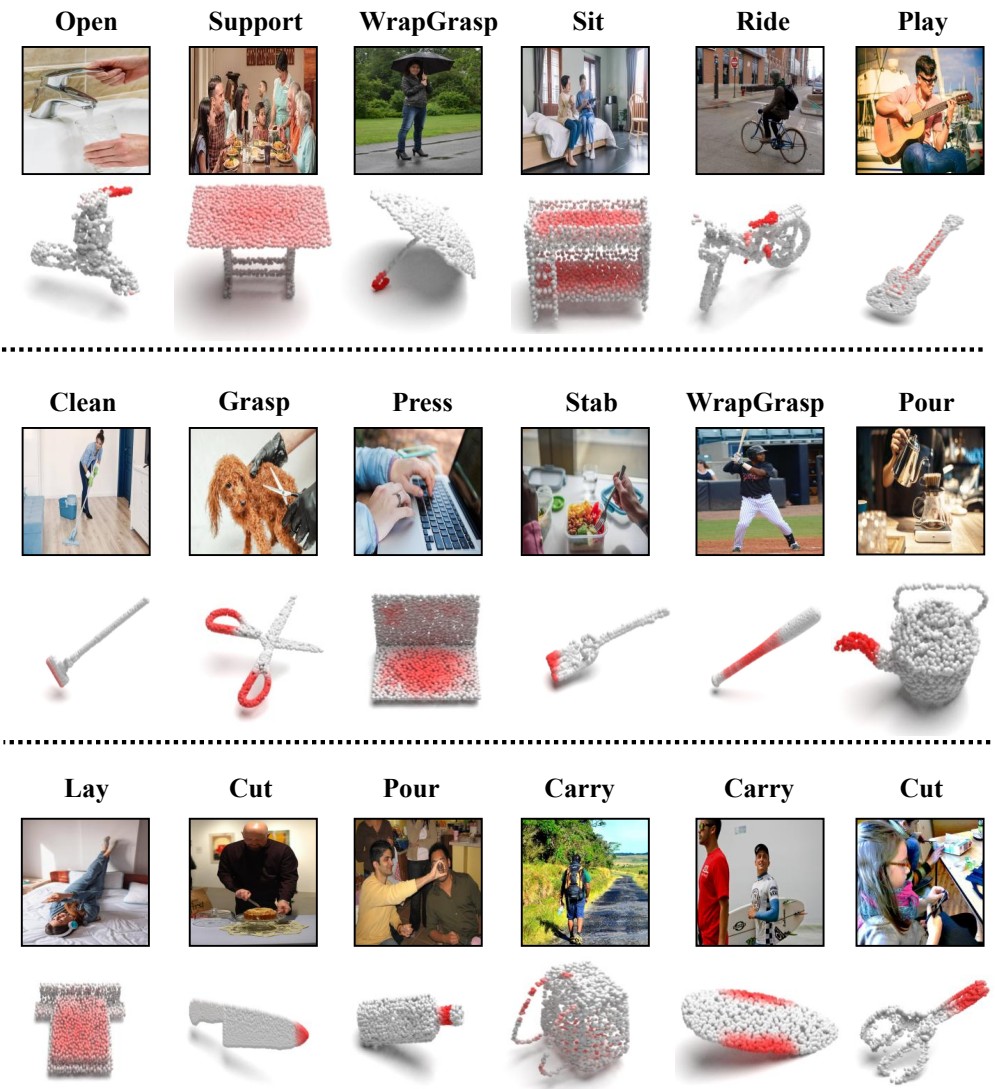

*Figure 6.* The first two rows are for the Seen Object setting, the middle two rows are for the Unseen Object setting, and the last four rows are for the Unseen Affordance setting.

