# OpenReview forum: "Learning Attribute–Affordance Hierarchies in Hyperbolic Space for Open-Vocabulary 3D Object Affordance Grounding"
_ICML.cc/2026/Conference — ICML 2026 regular_

### Official Review · Reviewer_fyd2 · 2026-02-27

**Soundness:** 3
**Presentation:** 3
**Significance:** 3
**Originality:** 2
**Overall Recommendation:** 4
**Confidence:** 4

**Summary:**

This work is in the field of Open-Vocabulary 3D Object Affordance Grounding (OVAG), attempting to localize action-relevant regions on 3D objects by interaction images or textual instructions. The authors propose the Attribute-Affordance Hierarchies (AAH) framework, which explicitly models the hierarchical relationship between local object attributes and affordances. The method combines three components: a hypergraph module to capture high-order local region relationships, hyperbolic space embedding to encode the attribute with affordance and 3D region hierarchy, and counterfactual attribute prompts to reduce spurious correlations. Experiments on the PIADv2 benchmark demonstrate good performance across seen, unseen-object, and unseen-affordance settings.

**Compliance With Llm Reviewing Policy:**

Affirmed.

**Final Justification:**

I thank the authors for their detailed rebuttal and the effort in conducting additional experiments. The backbone experiments with DGCNN and Point Transformer convincingly demonstrate that AAH's improvements are not dependent on the choice of 3D encoder, which was one of my primary concerns. The three-way ablation (Euclidean+Hierarchical, Hyperbolic+Standard, Hyperbolic+Hierarchical) is well-designed and provides clearer evidence for the necessity of hyperbolic geometry beyond the hierarchical loss alone. The explanations regarding the hierarchical prior acting as a soft regularizer and the role of the hypergraph module in capturing global dependencies are reasonable.
That said, I would still encourage the authors to consider the following points for the final version: (1) the Sim-to-Real discussion would benefit from at least a lightweight robustness experiment, such as adding Gaussian noise to point clouds, rather than relying solely on conceptual justification; (2) a qualitative failure analysis showing cases where the hierarchical prior is mismatched with the affordance structure would add intellectual honesty to the paper; and (3) the Unseen Affordance AUC of 80.11 for "Ours (Transformer)" appears notably higher than other entries in that column and may warrant a double-check. These are suggestions for improvement rather than outstanding objections.

**Key Questions For Authors:**

1. Have the authors evaluated whether AAH's performance gains hold when integrated with modern 3D encoders (e.g., Point Transformer)? Would the improvements be narrowed with stronger backbones?
2. Could the authors provide a direct comparison of "Euclidean Space + Hierarchical Loss" vs. "Hyperbolic Space + Hierarchical Loss"? If similar gains are achievable in Euclidean space, the necessity of hyperbolic geometry requires stronger justification.
3. Has the method been validated on real robotic platforms or with noisy sensor inputs? For affordance heat maps with boundary uncertainty, are there post-processing mechanisms to ensure safe physical interaction?
4. Could the authors clarify the rationality of this design and how semantic completeness is guaranteed across diverse affordance types? Is there a risk of conflict between this hierarchical inductive bias and the true structure of certain affordances? Also, how does the model handle affordances that depend on global object properties

**Limitations:**

yes

**Strengths And Weaknesses:**

**Strengths:**

For motivation and problem formulation, the insight that affordances emerge from specific attribute compositions rather than global appearance is well-founded and interesting. This may address a genuine limitation in prior visual-only alignment methods and suggest a potential approach for enhancing open-vocabulary generalization.

The evaluation on PIADv2 is comprehensive, covering three challenging settings (Seen, Unseen Object, Unseen Affordance) with multiple metrics (AUC, aIoU, SIM, MAE). Ablation studies (Table 2 & 3) provide some empirical evidence for each component's contribution.

The method has shown improvements over baselines (IAG, LASO, GREAT), particularly in Unseen Affordance settings where generalization is most critical.

**Weaknesses:**

The method relies on PointNet++ as the 3D encoder, which may not be the latest. However, this choice likely caps the achievable performance. Modern architectures (e.g., Point Transformer, DGCNN, or diffusion-based 3D encoders) offer significantly stronger geometric feature extraction. It should be further discussed whether AAH's benefits persist when integrated with stronger backbones.

The Ablation study part shows that the Euclidean+CL (Euclidean space + linear fusion) approach exhibits only a limited gap compared to the full hyperbolic model on certain metrics. The performance improvement likely stems primarily from the design of the loss function rather than from hyperbolic geometry itself. While hyperbolic space is theoretically well-suited to capturing hierarchical relations, the fusion mechanism is relatively basic, relying on simple integration of features.

The framework of this paper assumes a strict "attribute → affordance → 3D region" hierarchy (Figure 1). However, not all affordances conform to this structure. The paper needs to discuss how the model handles affordances that violate the hierarchical prior, nor does it analyze potential failure modes when the inductive bias is mismatched with the task.

The conclusion acknowledges scalability concerns but does not address the Sim-to-Real gap. PIADv2 contains synthetic and web-scraped data; affordance localization errors that are acceptable in benchmark metrics may lead to physical interaction failures in real robotic systems. The paper should discuss the potential performance falling on robustness to sensor noise, prediction uncertainty calibration, or safety-aware post-processing.

---

> ### Author Rebuttal · Authors · 2026-03-29
>
> 1. *The stronger backbones.*
>
> Thank you for your valuable suggestion. We have replaced the 3D encoder with Point Transformer and DGCNN for additional experiments. The results are presented in Table 1. **AAH continues to achieve consistent improvements over its corresponding baseline even under this stronger backbone**.
>
> *Table 1: More results on PIADv2.*
> |Methods|Seen AUC|Seen aIOU|Seen SIM|Seen MAE|Unseen Object AUC|Unseen Object aIOU|Unseen Object SIM|Unseen Object MAE|Unseen Affordance AUC|Unseen Affordance aIOU|Unseen Affordance SIM|Unseen Affordance MAE|
> |-|-|-|-|-|-|-|-|-|-|-|-|-|
> |GREAT (DGCNN)| 92.32| 38.22| 0.679| 0.065| 79.86| 20.42| 0.409| 0.108| 69.93| 12.07| 0.296| 0.125|
> |**Ours (DGCNN)** | **93.70**| **39.00** |**0.689** |**0.060** |**81.29**|**21.39**|**0.431**|**0.090**|**70.96**             |**13.30**|**0.326**|**0.112**|
> |GREAT (Transformer)| 92.91| 38.64| 0.682| 0.063| 80.15| 20.66| 0.412| 0.107| 70.24| 12.10| 0.299| 0.123|
> |**Ours (Transformer)** | **94.20**| **39.23** |**0.690** |**0.057** |**81.67**|**21.65**|**0.433**|**0.087**|**80.11**             |**13.46**|**0.328**|**0.111**|
> |Euclidean + Hierarchical| 92.75 | 38.55 | 0.680 | 0.063 | 80.40 | 20.45 | 0.411 | 0.099 | 70.25 | 12.35 | 0.304 | 0.121 |
> |Hyperbolic + Standard| 92.50| 38.35| 0.677| 0.064| 80.32| 20.36| 0.409| 0.101| 70.05| 12.20| 0.298| 0.123|
> | **Hyperbolic + Hierarchical (Ours)** | **93.29** | **38.98** | **0.688** | **0.060** | **81.01** | **21.18** | **0.430** | **0.090** | **70.90** | **13.26** | **0.320** | **0.113** |
>
>
>
> 2. *The comparsion of loss.*
>
> We thank the reviewer for this insightful suggestion.  We have conducted additional comparison. The results are shown in Table 1.  Adding hierarchical loss in Euclidean space improves the standard Euclidean baseline, indicating that the hierarchical supervision is indeed beneficial.
> The performance gain brought by this Euclidean Space + Hierarchical Loss is notably smaller than Hyperbolic Space + Hierarchical Loss, which highlights the necessity of modeling in hyperbolic space.
> This suggests that hierarchical loss alone is insufficient: while it encourages ordering, Euclidean space lacks the inductive bias to represent hierarchical relations without distortion. The hyperbolic geometry, with its radial organization and entailment structure, provides a more suitable space for encoding such semantic hierarchies.
>
> 3. *The Sim-to-Real gap.*
>
> We further explain how the proposed method can be effectively applied to real-world robotic systems. The training process involves learning a mapping between 2D interaction cues and 3D object regions, which is not restricted to specific object instances. To enable such cross-instance generalization, we maintain a hierarchica concept pairing between input images and point clouds during training. This design allows the model to capture the intrinsic relationship between local object attributes and affordances, thereby improving its generalization capability in real-world scenarios. To ensure safe physical interaction, we further design an adaptive thresholding strategy that converts continuous heatmaps into discrete interaction masks. Regions with high confidence scores but unstable geometric properties (e.g., insufficient contact area) are filtered out, while regions with lower uncertainty are prioritized as affordance grasping areas.
> This design provides a reliable and robust foundation for subsequent robotic control tasks.
>
> 4. *The discuss of the hierarchical prior*
>
> We thank the reviewer for this insightful comment.
> The hierarchy is not enforced as a hard constraint. The ordering and counterfactual losses act as regularizers in the embedding space, while the final prediction still relies on fused features with standard supervision. This allows the model to fall back to direct feature learning when an affordance does not conform well to the hierarchy, mitigating potential conflicts.
> The certain affordances—especially those depending on global object structure, multi-part interactions may not be well captured by this prior. However, our Hypergraph module is designed to capture non-local dependencies. By constructing hyperedges that span multiple regions, the model can aggregate global geometric statistics , which are crucial for affordances that depend on the object's total configuration.

---

> > ### Author Rebuttal · Reviewer_fyd2 · 2026-04-03
> >
> > I thank the authors for their detailed rebuttal and the effort in conducting additional experiments. The backbone experiments with DGCNN and Point Transformer convincingly demonstrate that AAH's improvements are not dependent on the choice of 3D encoder, which was one of my primary concerns. The three-way ablation (Euclidean+Hierarchical, Hyperbolic+Standard, Hyperbolic+Hierarchical) is well-designed and provides clearer evidence for the necessity of hyperbolic geometry beyond the hierarchical loss alone. The explanations regarding the hierarchical prior acting as a soft regularizer and the role of the hypergraph module in capturing global dependencies are reasonable.
> >
> > That said, I would still encourage the authors to consider the following points for the final version: (1) the Sim-to-Real discussion would benefit from at least a lightweight robustness experiment, such as adding Gaussian noise to point clouds, rather than relying solely on conceptual justification; (2) a qualitative failure analysis showing cases where the hierarchical prior is mismatched with the affordance structure would add intellectual honesty to the paper; and (3) the Unseen Affordance AUC of 80.11 for "Ours (Transformer)" appears notably higher than other entries in that column and may warrant a double-check. These are suggestions for improvement rather than outstanding objections.

---

> > > ### Author Response · Authors · 2026-04-06
> > >
> > > We sincerely thank the reviewer for the positive feedback and for acknowledging the additional experiments and clarifications. We will carefully revise the paper accordingly and further strengthen the final version based on your valuable feedback.

---

### Official Review · Reviewer_eYS2 · 2026-03-12

**Soundness:** 2
**Presentation:** 2
**Significance:** 1
**Originality:** 1
**Overall Recommendation:** 2
**Confidence:** 4

**Summary:**

This paper focused on the problem of open-vocabulary 3D object affordance grounding (OVAG), which aims to localize action-relevant regions on 3D objects using interaction images or textual descriptions. The authors noticed that existing approaches mainly rely on cross-modal alignment between images, text, and 3D geometry, but often overlook the intrinsic relationship between local object attributes and affordances, which limits localization accuracy and generalization to unseen affordances.

To solve this problem, the paper proposes an Attribute–Affordance Hierarchies (AAH) framework that explicitly models hierarchical relationships between object-region attributes and affordances. The method constructs a hypergraph to capture relationships among local visual regions, and embeds these features into a hyperbolic space to represent attribute–affordance hierarchies. It also introduces counterfactual attribute prompts to reduce spurious correlations between attributes and affordances. Experiments on the PIADv2 benchmark show improved performance for affordance localization, particularly in unseen object and unseen affordance scenarios.

**Compliance With Llm Reviewing Policy:**

Affirmed.

**Key Questions For Authors:**

How would the proposed framework extend to dexterous hand manipulation, where affordances depend on more complex contact geometry and finger coordination? Could the authors clarify how much improvement comes specifically from the hyperbolic hierarchical structure compared to simpler Euclidean relational models or transformer-based relational reasoning? How sensitive is the performance to the quality or diversity of these prompts, and how stable is the method if the prompts are noisy or incorrect?

**Limitations:**

yes

**Strengths And Weaknesses:**

Overall, the authors investigate an important concept by explicitly modeling the hierarchical relationship between object attributes and affordances rather than relying only on cross-modal alignment. The idea of representing attribute–affordance structures using hyperbolic space is conceptually well motivated for hierarchical data. The proposed AAH framework combines hypergraph modeling, hyperbolic embeddings, and counterfactual prompts to jointly capture local region relations and hierarchical semantics, which is a coherent design. Experiments on PIADv2 show consistent improvements over previous methods across seen and unseen settings, suggesting better generalization. The paper includes ablation studies on hypergraph construction, loss terms, and hyperbolic embeddings, helping validate the contributions of different components. Overall, a major challenge discussed by this article is open-vocabulary affordance grounding, but the experiments mainly involve general object–gripper interactions rather than dexterous hand manipulation, leaving a noticeable gap between the problem setting and dexterous hand scenarios. The method combines multiple components (hypergraph construction, hyperbolic embedding, counterfactual prompting), making the pipeline relatively complex and potentially difficult to reproduce or deploy in real-time robotic systems. The framework relies on prompt generation and multimodal language reasoning to construct attribute–affordance relations, which may introduce noise or bias when prompts are ambiguous. While the method improves affordance localization benchmarks, the paper provides limited discussion or evaluation on downstream robotic manipulation tasks.

---

> ### Author Rebuttal · Authors · 2026-03-31
>
> We thank the reviewer for these insightful questions and suggestions.
>
> 1. *Dexterous manipulation.*
>
> Our current framework focuses on object-centric, visually grounded affordances. We agree that extending to dexterous hand scenarios requires modeling fine-grained contact geometry and coordinated multi-finger interactions. In our formulation, this can be naturally incorporated by (i) replacing region-level tokens with contact-point or surface-patch tokens, and (ii) introducing hand–object relational reasoning modules (e.g., graph or transformer-based interactions over fingers and contact regions). The current hierarchical design, attribute → affordance → region, can be extended to a richer structure such as ''attribute → affordance → contact pattern'', where contact patterns encode grasp type or finger coordination.
>
> 2. *Contribution of hyperbolic hierarchy.*
>
> To disentangle the effect of geometry and supervision, we conducted additional controlled comparisons (Euclidean + hierarchical loss vs. Hyperbolic + hierarchical loss) on Table 1. While hierarchical supervision improves Euclidean models, the hyperbolic variant consistently achieves better performance. This suggests that hyperbolic geometry provides a more suitable structure for modeling attribute-affordance-3d regions relations, rather than the gain being solely attributable to stronger loss design or feature fusion.
>
> *Table 1: More results on PIADv2.*
> |Methods|Seen AUC|Seen aIOU|Seen SIM|Seen MAE|Unseen Object AUC|Unseen Object aIOU|Unseen Object SIM|Unseen Object MAE|Unseen Affordance AUC|Unseen Affordance aIOU|Unseen Affordance SIM|Unseen Affordance MAE|
> |-|-|-|-|-|-|-|-|-|-|-|-|-|
> |Euclidean + Hierarchical| 92.75 | 38.55 | 0.680 | 0.063 | 80.40 | 20.45 | 0.411 | 0.099 | 70.25 | 12.35 | 0.304 | 0.121 |
> |Hyperbolic + Standard| 92.50| 38.35| 0.677| 0.064| 80.32| 20.36| 0.409| 0.101| 70.05| 12.20| 0.298| 0.123|
> | **Hyperbolic + Hierarchical (Ours)** | **93.29** | **38.98** | **0.688** | **0.060** | **81.01** | **21.18** | **0.430** | **0.090** | **70.90** | **13.26** | **0.320** | **0.113** |
>
>
> 3. *Prompt sensitivity and robustness.*
>
> To investigate the robustness of our method to the quality of prompt information, we conduct noise injection experiments with noise ratios of 12.5% and 25%. As shown in Table 2, we observe that the model does not suffer from significant performance degradation and maintains competitive accuracy even under high noise conditions. We attribute this robustness to the joint effect of the counterfactual exclusion loss.
> This training strategy teaches the model to decouple affordance from auxiliary information. Consequently, at inference time, the model naturally learns to ignore irrelevant or noisy parts of the prompt that do not contribute to the essential causal chain of the affordance.
>
>
> *Table 2: Noise robustness of our method on PIADv2.*
> |Noise Rate|Seen AUC|Seen aIOU|Seen SIM|Seen MAE|Unseen Object AUC|Unseen Object aIOU|Unseen Object SIM|Unseen Object MAE|Unseen Affordance AUC|Unseen Affordance aIOU|Unseen Affordance SIM|Unseen Affordance MAE|
> |-|-|-|-|-|-|-|-|-|-|-|-|-|
> | 0% | **93.29**  | **38.98**   | **0.688**  | **0.060**  | **81.01**  | **21.18** | **0.430**| **0.090** | **70.90**  | **13.26** | **0.320**              | **0.113** |
> | 12.5%| 93.02 | 38.71 | 0.683 | 0.061| 80.63| 20.86| 0.423 | 0.093 | 70.41| 12.98| 0.312 | 0.117 |
> | 25% | 92.63 | 38.32| 0.676| 0.064| 80.14| 20.44| 0.414| 0.097 | 69.82 | 12.61| 0.303| 0.121 |

---

> > ### Author Rebuttal · Reviewer_eYS2 · 2026-04-03
> >
> > Thanks authors for rebuttal.
> > My question is partially solved but I have much concern on the necessties of this work because for rigid body affordance. It can be simply detected with part segementation and LLM. The value of affordance not lies in point choose but on orientation how in this case the small trajectory of manipulation ie the trajectory of lift or support vector. Thus I ramain my score.

---

> > > ### Author Response · Authors · 2026-04-08
> > >
> > > We thank the reviewer for the thoughtful comment.
> > >
> > > We agree that for some rigid objects, part segmentation + LLM can provide a reasonable method for identifying candidate regions. However, our work targets a different and more general problem: open-vocabulary 3D object affordance grounding, where (i) objects and affordances may be unseen, and (ii) grounding must be aligned with action regions rather than manipulation trajectory.
> > >
> > > Compared to part segmentation, segmentation does not explicitly encode which properties make a region actionable, while our framework learns this mapping in a structured manner. Our method models attribute–affordance relations (e.g., flat, hard→ support), enabling generalization beyond predefined parts and supporting compositional reasoning across objects.
> > > Regarding manipulation, we fully agree that orientation and trajectories are critical. Our current work focuses on perception-level affordance grounding, which can serve as a semantic prior for downstream modules. The predicted regions and their associated attributes can be directly used to infer grasp directions, or support vectors, which are then refined by motion planning or control policies.
> > >
> > > We will further extend our framework to dexterous hand manipulation in the revised version.

---

### Official Review · Reviewer_9RcE · 2026-03-13

**Soundness:** 4
**Presentation:** 4
**Significance:** 4
**Originality:** 4
**Overall Recommendation:** 4
**Confidence:** 4

**Summary:**

This paper studies open-vocabulary 3D object affordance grounding, with the goal of localizing affordance regions on 3D objects using interaction images and textual prompts. The main idea is to model affordances as emerging from local object attributes, and to encode attribute-affordance hierarchies using a combination of hypergraph-based local relation modeling and hyperbolic embeddings. The method also introduces counterfactual attribute prompts to reduce spurious attribute-affordance correlations. Experiments on PIADv2 show improvements over prior baselines in seen, unseen-object, and unseen-affordance settings, with ablations on the hypergraph, losses, and hyperbolic space.

**Compliance With Llm Reviewing Policy:**

Affirmed.

**Key Questions For Authors:**

Can you provide direct evidence that the learned hyperbolic embedding captures the intended attribute-affordance hierarchy, beyond downstream performance?

**Limitations:**

yes

**Strengths And Weaknesses:**

Strengths

1. This paper is well-motivated. Open-vocabulary 3D affordance grounding is a meaningful task for multimodal learning and embodied AI.

2. The paper is coherent, well-written and easy to following.

3. The method is reasonable. Figure 2 gives a useful overview of how the image branch, prompt branch, hyperbolic embedding, and point-cloud branch interact. In particular, the figure makes clear that the method is not just “hyperbolic loss slapped onto a standard model,” but a full multimodal pipeline with explicit counterfactual prompting and hierarchy-aware objectives.

4. The ablations are directionally informative.

Weaknesses

1. The description of the method part has too many notation and exposition issues for comfort. For example, in Section 4.3, the notation for image and point embeddings is inconsistent, and the text around Equation (11) appears to confuse attribute and affordance embeddings. Equation (13) is particularly hard to parse, with nested functions and shape manipulations described in prose rather than clearly defined operators.

2. The computational cost is ignored despite being relevant. The method adds hypergraph construction, hyperbolic mapping, entailment-cone losses, and prompt processing. The conclusion itself admits possible scalability issues, yet the experiments report no runtime, memory, or training-cost numbers. This matters for practical adoption and for judging whether the gains are worth the added complexity.

---

> ### Author Rebuttal · Authors · 2026-03-30
>
> 1. *The description of the method part has too many notation and exposition issues for comfort.*
>
> We thank the reviewer for the careful reading.
> In the future revision, we will introduce a unified notation scheme that clearly separates image and point embeddings.
> For Eq. (11),  we will rewrite this part as:
>
> $\begin{aligned} & L_{\mathrm{ord}}^{a \rightarrow o}=\max \left(0, \theta\left(h_a^I, h_o^I\right)-\phi\left(h_a^I\right)\right), \\ & L_{\mathrm{ord}}^{o \rightarrow P}=\max \left(0, \theta\left(h_o^I, h^P\right)-\phi\left(h_o^I\right)\right) .\end{aligned}$
>
> For Eq. (13), we will replace it with a sequence of explicitly defined operators:
> ''To confuse $T_o \in \mathbb{R}^{N_o \times C}$ with the point cloud feature,
> we first choice $f_\Theta$, which expand $T_o$ to $\mathbb{R}^{N_p \times C}$. Then, we concatenate it with $F_p$ and use $f$, which indicates convolution layers with a 1×1 kernel, to encode them.
> After we get the fused feature $\bar{F}_p$, we follow the point cloud upsample operation to change the $\bar{F}_p$ to $\mathbb{R}^{C \times N}$ by Feature Propagation layers.''
>
> ''$f_\Gamma$ reshapes the fused image features $F_a$ to $\mathbb{R}^{C \times N}$.  And the final output ${F}_{\alpha} \in \mathbb{R}^{C \times N}$ is affordance feature representation.''
>
> 2. *The computational cost is ignored despite being relevant.*
>
> We thank the reviewer for pointing out the importance of computational cost.
> In the revision, we will report model inference time, model parameters and trainable parameters.
> We have calculated them as shown in the following table:
>
> *Table 1: Statistics of Time and Param.*
> | **Model** | **#Time** |**#Params.** | **#Trainable Params.** |
> |:-----:|:-----:|:-----:|-----:|
> |  IAG  |  1.426s | 24.7M | 24.7M |
> |  LASO  |  1.336s | 130.5M | 130.5M |
> |  GREAT  |  1.272s | 256.7M | 23.1 M |
> |  Ours  |  2.1s | 415.2M | 150M |
>
> The results show that our method introduces a moderate increase in both parameter count and inference time compared to the strongest baseline. This overhead mainly comes from the hypergraph module and the additional hyperbolic embedding and loss computations. However, these components operate on compact feature representations, and the overall increase remains limited relative to the backbone cost.
>
> 3. *The evidence of attribute-affordance hierarchy.*
>
> We thank the reviewer for this important suggestion.
> To further verify that the hyperbolic structure captures the intended attribute–affordance–3D region hierarchy, we present in Fig. 1 a two-dimensional visualization of the learned hyperbolic embeddings using UMAP. In the figure, colors denote categories; the largest points represent attributes, the medium-sized points denote affordances, and the small points correspond to 3D regions. Beyond the clear clustering pattern induced by category labels, the visualization also reveals an intriguing attribute–affordance–3D point-cloud hierarchy, where attributes tend to lie closer to the center of the disk, affordances are distributed around them, and 3D point clouds are located toward the outer boundary.
>
> Fig.1 anonymous link: https://imgur.com/a/jYXn5PS

---

> > ### Author Rebuttal · Reviewer_9RcE · 2026-04-07
> >
> > Thanks for the detailed rebuttal.

---

> > > ### Author Response · Authors · 2026-04-08
> > >
> > > We sincerely thank the reviewer for the positive feedback. We will carefully revise the paper accordingly and further strengthen the final version based on your valuable feedback.

---

### Official Review · Reviewer_ZmYf · 2026-03-17

**Soundness:** 3
**Presentation:** 3
**Significance:** 1
**Originality:** 2
**Overall Recommendation:** 3
**Confidence:** 5

**Summary:**

The paper studies open-vocabulary 3D object affordance grounding (OVAG), where the goal is to localize action-relevant regions on 3D objects using interaction images or text. The authors propose AAH (Attribute-Affordance Hierarchies), which first uses a hypergraph to model local relations among visual components and then maps region-level representations into hyperbolic space to encode abstract-to-specific attribute–affordance hierarchies. The method also introduces counterfactual attribute prompts to perturb attribute conditions and encourage the model to learn more robust attribute–affordance dependencies rather than relying on spurious correlations. On PIADv2, AAH reports the best results across Seen, Unseen Object, and Unseen Affordance splits, improving over prior baselines such as GREAT. Overall, the paper’s contribution is to show that explicitly modeling hierarchical attribute-to-affordance structure with hypergraph reasoning, hyperbolic embeddings, and counterfactual supervision can improve open-vocabulary 3D affordance localization.

**Compliance With Llm Reviewing Policy:**

Affirmed.

**Final Justification:**

Thanks again to the authors for the rebuttal and additional experiments. That said, as I explained, my main concern about the cost-to-gain trade-off remains. I wouldn't object to acceptance, but overall I'm borderline, leaning slightly negative.

**Key Questions For Authors:**

* The paper would benefit from more direct evidence that the proposed hyperbolic attribute–affordance space is genuinely better structured than the baseline GREAT representation, e.g., through visualization or probing of representative examples rather than relying mainly on small performance gains.

* Was the GREAT (DINO) baseline fine-tuned after changing the image embedding?

**Limitations:**

Missing reference:

Lee et al., 2025. Affogato: Learning Open-Vocabulary Affordance Grounding with Automated Data Generation at Scale

Minor:

Sec 4.1 title: hpergraph → hypergraph

L367 and L368 are repetitive.

**Strengths And Weaknesses:**

Strengths:

- Clear writing, easy to follow.
- Well motivated: explicitly modeling attribute–affordance relations is a reasonable way to improve generalization beyond direct cross-modal alignment.
- The method shows consistent gains across all three evaluation splits on PIADv2.

Weaknesses

- The improvement over the strongest baseline (GREAT) is fairly modest. The gains are consistent but small in absolute terms: for example, the AUC improvement is about +1.3 / +1.4 / +1.1 across Seen / Unseen Object / Unseen Affordance, and the SIM gains are also relatively limited. This makes the overall advance feel incremental rather than substantial. The method appears architecturally heavy relative to the empirical gain.
- The ablation suggests that the main improvement may come from the contrastive loss rather than the proposed hierarchical/hyperbolic design itself. In Table 2, removing $L_{cont}$ causes the largest drop, whereas removing the hypergraph or replacing the hyperbolic design leads to noticeably smaller changes. This raises the question of whether a stronger GREAT-style baseline with the same contrastive objective could recover much of the reported gain.

---

> ### Author Rebuttal · Authors · 2026-03-29
>
> We thank the reviewer for this important comment.
> 1. *The improvement over the strongest baseline (GREAT) is fairly modest.*
>
> The design of our method is intended to focus on the intrinsic relationship between local object attributes and affordances, rather than merely aligning attribute and affordance knowledge with 2D and 3D visual representations. Our method demonstrates consistently improved performance across all evaluation metrics. Notably, these gains are particularly significant under the “unseen objects and unseen affordances” setting. Furthermore, the effectiveness of our approach is strongly supported by ablation studies: removing any component of the proposed framework leads to performance degradation to varying degrees. In particular, when the hyperbolic space design is replaced with its Euclidean counterpart, the model exhibits a consistent drop in performance across all evaluation splits. This observation provides strong evidence that the performance improvements stem from the proposed hierarchical modeling mechanism. In addition, we compare our method with several existing approaches in terms of parameter count and computational cost, as shown in Table 1. The results indicate that our method does not incur excessive model complexity.
>
> *Table 1: Statistics of Time and Param.*
> | **Model** | **#Time** |**#Params.** | **#Trainable Params.** |
> |:-----:|:-----:|:-----:|-----:|
> |  IAG  |  1.426s | 24.7M | 24.7M |
> |  LASO  |  1.336s | 130.5M | 130.5M |
> |  GREAT  |  1.272s | 256.7M | 23.1 M |
> |  Ours  |  2.1s | 415.2M | 150M |
>
> 2. *The same contrastive objective.*
>
> Thank you for the valuable suggestion. The contrastive loss term is primarily responsible for aligning matched image/point cloud instances. In contrast, the hyperbolic formulation further enforces semantic ordering through $L_{ord}$ and enhances the clarity of hierarchical boundaries via $L_{cf}$.
> To further investigate this, we incorporate an InfoNCE-style contrastive objective into the GREAT model for evaluation. The results, as shown in Table 2, indicate that the performance improvement is marginal.
>
> *Table 2: Comparsion on PIADv2.*
> | Methods   | Seen AUC | Seen aIOU | Seen SIM | Seen MAE | Unseen Object AUC | Unseen Object aIOU | Unseen Object SIM | Unseen Object MAE | Unseen Affordance AUC | Unseen Affordance aIOU | Unseen Affordance SIM | Unseen Affordance MAE |
> |-|-|-|-|-|-|-|-|-|-|-|-|-|
> |GREAT| 91.99| 38.03| 0.676| 0.067| 79.57| 20.16| 0.402| 0.109| 69.81| 12.05| 0.290| 0.127|
> | **GREAT+$L_{cont}$**  | **92.34**| **38.20** |**0.680** |**0.064** |**79.99**|**20.25**|**0.411**|**0.105**|**70.10**             |**12.20**|**0.295**|**0.125**|
>
> 3. *More direct evidence.*
>
> We thank the reviewer for this valuable suggestion. We perform a UMAP projection for both our method and the GREAT baseline. In the Fig 1, large and medium-sized points denote attribute and affordance embeddings, respectively, while small points correspond to 3D region features.
> Compared to GREAT, our method exhibits a noticeably clearer hierarchical organization. In particular, attribute embeddings tend to cluster more centrally, affordances occupy intermediate regions, and 3D region features are distributed toward the periphery, forming a consistent coarse-to-fine structure.
> In contrast, the GREAT representation shows less structured organization and weaker separation between levels.
>
> 4. *The GREAT (DINO) baseline.*
>
> Following the GREAT setting, we have frozen the DINO baseline during training.
>
> Fig. 1 anonymous link: https://imgur.com/a/jYXn5PS

---

> > ### Author Rebuttal · Reviewer_ZmYf · 2026-04-04
> >
> > I appreciate the additional GREAT+Lcont experiment and the UMAP visualization, which together provide stronger evidence that the gains are not solely attributable to the contrastive loss. However, the cost-to-gain ratio remains a concern: 1.6× the parameters and 1.65× the inference time for ~1 point AUC improvement does not, in my view, constitute a sufficiently compelling advance. I maintain my score.

---

> > > ### Author Response · Authors · 2026-04-08
> > >
> > > We thank the reviewer for the detailed follow-up and for acknowledging our additional evidence.
> > >
> > > Our goal is to improve structured representation and generalization in open-vocabulary affordance grounding. Beyond AUC, our method consistently improves across multiple metrics (SIM, MAE, aIOU) and, more importantly, demonstrates stronger generalization and hierarchical organization (as supported by visualization results).
> > > These aspects are not captured by AUC alone but are critical for open-vocabulary and downstream reasoning tasks.
> > >
> > > Furthermore, the additional parameters in our method are not for "brute-force" performance boosting but for encoding attribute-affordance structure. By introducing Hyperbolic Hierarchical Modeling and Hypergraph Construction, our model moves beyond simple visual-semantic alignment (as in GREAT) to explicitly modeling the "Why" and "How" of affordances, which captures the intrinsic relationship between local object attributes and affordances.
> > >
> > > Our work provides a new paradigm for OVAG—modeling attribute-affordance hierarchies—rather than a simple incremental tweak. We hope the reviewer considers the value of this structural advancement alongside all metrics.

---

### Decision · Program_Chairs · 2026-04-30

**Decision:**

Accept (regular)

**Comment:**

This paper received mixed reviews: 3, 4, 2, 4.

The reviewers note that the work is "well-motivated" (9RcE), with a "comprehensive" evaluation on PIADv2 (fyd2). The review of fyd2 (4) makes a convincing positive case for the work overall, particularly based on the ablations added in the rebuttal on encoder types and the three-way ablation (euclidean/hyperbolic/standard/hierarchical) which lend credence to the effectiveness of the paper's main idea.

Some concerns remain, regarding the method's complexity and practical applicability. ZmYf notes that the 'cost-to-gain ratio remains a concern: 1.6× the parameters and 1.65× the inference time for ~1 point AUC improvement does not, in my view, constitute a sufficiently compelling advance.' 9RcE (4) similarly notes that the method introduces substantial architectural complexity.

The most negative review, from eYS2 (2), focuses primarily on dexterous manipulation by robots, which is relevant but not really the focus of the paper.

Taking this in sum, the AC finds this paper imperfect but acceptable. The AC highlights that fyd2 makes specific constructive recommendations on how to improve the paper, and 9RcE also notes 'too many notation and exposition issues for comfort' -- the AC agrees, but hopes these revisions are feasible for the authors. Congratulations on the acceptance, and please work hard to improve the paper for camera-ready, along the lines recommended by fyd2 and 9RcE.